# The Usefulness of a Duplex RT-qPCR during the Recent Yellow Fever Brazilian Epidemic: Surveillance of Vaccine Adverse Events, Epizootics and Vectors

**DOI:** 10.3390/pathogens10060693

**Published:** 2021-06-03

**Authors:** Alice L. N. Queiroz, Rafael S. Barros, Sandro P. Silva, Daniela S. G. Rodrigues, Ana C. R. Cruz, Flávia B. dos Santos, Pedro F. C. Vasconcelos, Robert B. Tesh, Bruno T. D. Nunes, Daniele B. A. Medeiros

**Affiliations:** 1Department of Arbovirology and Haemorrhagic Fevers, Evandro Chagas Institute, Ananindeua 67030-000, Brazil; rafaelsantos2829@gmail.com (R.S.B.); spatroca@gmail.com (S.P.S.); danielas.g.rodrigues@hotmail.com (D.S.G.R.); anacecilia@iec.gov.br (A.C.R.C.); pedro.vasconcelos@globo.com (P.F.C.V.); danielemedeiros@iec.gov.br (D.B.A.M.); 2Viral Immunology Laboratory, Oswaldo Cruz Institute, Rio de Janeiro 21040-900, Brazil; flaviab@ioc.fiocruz.br; 3Department of Pathology and Microbiology & Immunology, University Texas Medical Branch, Galveston, TX 77555, USA; rbtesh@gmail.com

**Keywords:** yellow fever, duplex RT-qPCR, vaccine adverse effects, Brazil

## Abstract

From 2016 to 2018, Brazil faced the biggest yellow fever (YF) outbreak in the last 80 years, representing a risk of YF reurbanization, especially in megacities. Along with this challenge, the mass administration of the fractionated YF vaccine dose in a naïve population brought another concern: the possibility to increase YF adverse events associated with viscerotropic (YEL-AVD) or neurological disease (YEL-AND). For this reason, we developed a quantitative real time RT-PCR (RT-qPCR) assay based on a duplex TaqMan protocol to distinguish broad-spectrum infections caused by wild-type yellow fever virus (YFV) strain from adverse events following immunization (AEFI) by 17DD strain during the vaccination campaign used to contain this outbreak. A rapid and more accurate RT-qPCR assay to diagnose YFV was established, being able to detect even different YFV genotypes and geographic strains that circulate in Central and South America. Moreover, after testing around 1400 samples from human cases, non-human primates and mosquitoes, we detected just two YEL-AVD cases, confirmed by sequencing, during the massive vaccination in Brazilian Southeast region, showing lower incidence than AEFI as expected.

## 1. Introduction

Yellow fever (YF) caused by the yellow fever virus (YFV), an arbovirus member of the Flavivirus genus and Flaviviridae family, is an endemic zoonotic disease in tropical regions of Africa and South America [1,2]. In the latter, two genotypes with distinct lineages—South America (SA) I and II—have been reported, while in Africa five genotypes have been identified [3].

YFV human infection produces clinical manifestations ranging from self-limited to life-threatening disease. Symptomatic cases in general begin suddenly with high fever, nausea, vomiting and myalgias followed by remission and recovery. About 15% of cases do not recover and progress to an intoxication phase with high fever, multisystem organ failure and eventual death [4,5].

Although the majority of human cases are not reported, the actual global burden of YF is estimated to be around 200,000 cases per year, of which near 90% have been reported in Africa where the case fatality rate (CFR) is 20% lower than the CRF reported in SA (40%) [6]. 

YFV is maintained in Nature by two main transmission cycles. The sylvatic cycle involves non-human primates (NHP) and jungle mosquitoes, such as *Haemagogus* spp. and *Sabethes* spp. (in the Americas) and *Aedes africanus* and other sylvatic *Aedes* mosquito species (in Africa). The urban cycle involves *Aedes aegypti* and humans [7]. Despite the availability of a safe and highly efficient vaccine for more than 85 years, an increased number of YF outbreaks has recently been reported in Africa (Republic of the Congo, Angola, Uganda and Nigeria) and Brazil [8,9].

In Brazil, the last case of urban YF was reported in 1942 and since then YFV have been restricted to forest and rural areas with epizootic and occasional sporadic human outbreaks [10]. Historically, most YF cases in the country have been reported in the hydrographic basins of the Amazon, Araguaia, Tocantins, and Parana rivers [11]. However, this changed in the last few decades, with the YFV detection now occurring outside the limits of the endemic Amazon region.

In 2008, a remarkable spread of YF was observed, with cases initially reported in the North region (Pará and Tocantins), Central Region (Goiás and Mato Grosso do Sul) and South Region (Rio Grande do Sul). In 2009, São Paulo, a state so far considered free of virus circulation and thus with no recommendation for vaccination, reported a CFR of 39% [4]. Between 2016–2017, the most significant YF outbreak in the history of Brazil was reported, occurring mainly in Southeast region with 1376 cases and 483 deaths (CFR 35.1%) confirmed in the states of Minas Gerais, São Paulo, Rio de Janeiro, Espírito Santo and in the Federal District [12].

The risk of outbreak is primarily controlled via vaccination coverage of vulnerable human populations [13]. In Brazil, the vaccination programs have long been restricted to the North and Central regions. However, with the 2016 YF outbreak approaching the international megacities of Brazil’s Southeast coast [14], a massive vaccination campaign was implemented to contain the outbreak, with the production and delivery of millions of vaccine doses in the affected areas. Since adverse vaccine effects have been reported, associated with either viscerotropic disease (YEL-AVD), which presents undistinguished signs and symptoms from the natural infection, or with neurological disease (YEL-AND) [15], there was some concern that the increase in vaccination would lead to a increase in vaccine adverse events. 

In this respect, during the 2016–2017 outbreak, nearly 26 million YF vaccine doses were allocated to 1050 municipalities in Southeast region [13,16]. A total of 2068 suspected cases of vaccine adverse effects were reported, an incidence of 0.008% percent per 100,000 dose [17].

Here, we first aimed to establish a rapid duplex TaqMan RT-qPCR- assay able to differentiate natural circulating YFV American Genotypes strains from the YFV 17DD vaccine strain. Second, we aimed to use this newly developed duplex assay to investigate YEL-AVD cases. Finally, we performed virologic surveillance in mosquitoes and NHP involved in the Brazilian 2016–2017 outbreak. The assay developed in this study could become a valuable tool when used in the Public Health System to promptly characterize the YFV circulation and identify cases of severe adverse events caused by 17DD vaccine during outbreaks.

## 2. Results

### 2.1. YFV Duplex Taqman RT-qPCR Assay Standardization

In this study we aimed to develop a duplex qRT-PCR assay able to detect and distinguish between wildtype YFV American Genotypes strains and YFV 17DD vaccine strain. For this, we designed and optimized primers and probes that allowed a simultaneous detection of both targets in a single tube with high efficiency.

The bioinformatics analysis did not reveal any primer dimer or harpin structures for YFall and 17DD primers and probes. The best primers and probe concentrations were of 400 nM for each primer (forward and reverse) and 200 nM for each TaqMan probe, using the Quantitec Probe RT-qPCR Master Mix. The overall duplex format performance was similar to the singleplex format performance, with an EF% (efficiency) average equal to 100.55%. Samples with CT values lower than 38 were considered as positive. This is replicated in Figure 1.

In order to evaluate the assay linear dynamic range (LDR), defined as the range of target concentration that may be amplified with acceptable linearity, a serial dilution ranging from 10^2^ to 10^7^ copies/reaction of in vitro transcribed (IVT) YFV RNA was tested. The LDR assay was determined at 10^3^ to 10^7^ copies (R^2^ = 0.99), with a limit of quantification (LOQ) of 10^3^ copies/reaction. The endpoint limit of detection (LOD) of the assay was 10^2^ copies/reaction (Ct: 36.2 for YFall and 37.1 for 17DD). This is summarized in Table 1.

The duplex assay usefulness was also assessed using the internal control in distinct clinical specimens, such as blood, serum and tissues from humans and NHP. Regardless of the type of sample, all internal controls showed satisfactory detection with minimal Ct variation.

To evaluate the assay analytic specificity, representative strains of other arboviruses circulating in Brazil: flaviviruses (Dengue 1 to 4, Zika virus, Ilheus virus, Saint Louis Encephalitis virus, Rocio virus); orthobunyaviruses (Catu virus, Caraparu virus, Tacaiuma virus, Icoaraci virus, Utinga virus, Oropouche virus and Jatobal virus); and togaviruses (Chikungunya virus, Mayaro virus, East Equine Encephalitis virus, West Equine Encephalis virus, Pixuna virus and Mucambo virus) were tested and both (YFall and 17DD/African) primers and probes established here were unable to detect and amplify their genomes.

The assay sensitivity was further evaluated using a total of 28 YFV strains isolated in six Latin American countries over 40 years (Appendix A). All samples (100%, 28/28) were positive by the duplex RT-qPCR assay and four of those (14.3%; 4/28) were amplified by 17DD/African primers and probes. This is illustrated in Figure 2.

### 2.2. Duplex Taqman RT-qPCR Assay Clinical Evaluation

The duplex RT-qPCR performance was compared to the gold standard method (virus isolation) and a conventional molecular assay routinely used for YFV detection. Clinical samples (*n* = 53) from 30 YF suspect cases were randomly selected and analyzed, including seven samples from two YEL-AND cases from previous outbreak (Goiás state, and Rio Grande do Sul state—GOI4191 e RS21) [18]. The viral isolation method identified YFV in 20.7% (11/53) of the samples tested. The conventional RT-PCR, was positive in 24.5% (13/53). However, a significant increase in positivity of approximately 4-fold (94.3%; 50/53) was observed when the same samples were submitted to the Duplex RT-qPCR protocol established here (Figure 2). In fact, this result was due the low analytic sensitivity of conventional RT-PCR, which presented a LOD much higher than the RT-qPCR (10^11^ copies/reaction).

### 2.3. Investigation of YF Cases, Vaccine Adverse Events, Epizootics and Vectors during the 2016–2017 Epidemic

Human cases (*n* = 319), NHP cases (*n* = 512) and mosquito pools (*n* = 41) obtained from the 2016–2017 YF epidemic in Brazil were tested, using the duplex RT-qPCR. YFV was identified in 21.9% (70/319) of the human cases, 18.4% (93/512) of NHP and 19.5% (8/41) of mosquito pools. 

Most human YF cases originated from Minas Gerais (50/70), Espírito Santo (6/70) and Pará (8/70). The samples were collected between January and May 2017, during the massive YFV vaccine campaign in the Southeast region. This is shown in Figure 3.

In human specimens, YFV was identified by the duplex RT-qPCR in 8.9% (16/180) of serum tested, 50.0% (6/12) of the blood, 45.3% (53/117) of liver, 69.4% (50/72) of spleen, 26.3 (5/19) and 48.2% (40/83) of the pool of tissues. The virus was detected in 4.3% (2/46) of serum samples, 7.0% (3/43) of blood, 27.0% (85/315) of liver, 22.7% (77/319) of spleen, 26.8% (22/82) of brain and 10.5% (13/124) of the pool of tissues. These findings are presented in Table 2. 

The liver had a lower Ct average (18.6) than compared to the other specimens. Between 949 NHP specimens tested, YFV was detected in 21.3% (202/947). As observed for human specimens, the NHP liver had a lower Ct average (13.3) when compared to the other specimens. Using the duplex RT-qPCR, YFV was detected in 19.5% (8/41) of the mosquito pools tested and all positive pools were from *Haemagogus* species, presenting, on average, a Ct of 32.53. This is replicated in Figure 4.

From the YF human confirmed cases (*n* = 70), 97.2% (68/70) were positive for YFV WT genome detection. Two cases (2/70; 2.8%) from Espírito Santo (BeH846951, Ct = 29.7 and BeH849031, Ct = 29.9) were positive using the duplex RT-qPCR 17DD/African strain primer/probe, suggesting an adverse vaccine event. Both patients were vaccinated 3 days and 2 days, respectively, prior to the onset of the YF clinical manifestations. The first case was from a 67 years old female, living in a rural area. The second was a 30 years old male, living in the urban area. The signs and symptoms presented by both cases were similar to classic hemorrhagic YF and included hematemesis, melena, epistaxis, gum bleeding, abdominal pain, Faget signs (high temperature accompanied by slow pulse rate) and renal excretion disturbances. Despite all these symptoms, both patients recovered.

After virus isolation in cell culture, the strain BeH846951 was fully sequenced and the infection status by the 17DD vaccine subtype confirmed, with a nucleotide identity of 100% when compared to the 17DD vaccine strain (GenBank: DQ100292) and clustering with other vaccine strains (Appendix A). 

From the BeH849031 isolate, four sequence reads compatible to YFV were obtained, two spanning the capsid (201 nucleotides [nt]) and two the NS5 (255 nt) of the YFV 17DD/West African strain. However, it was not possible to distinguish between the vaccine strains from the sylvatic YFV West African genotype.

### 2.4. Phylogenetic Analysis

By either characterizing true positive samples used for the assay evaluation or to confirm the vaccine adverse events overall, this study determined 31 novel sequences of YFV nearly full genomes from Brazil, Ecuador, the USA, Venezuela, Peru, and Trinidad and Tobago collected between 1968 and 2011 (Appendix A). The results obtained from all recovered ORF with 10,236 nt.

In the phylogenetic tree, the samples were distributed into Three YFV genotypes; South America I (SAI) and South America II (SAII) of the America cluster, and Vaccine group related Asibi strain into West Central Africa cluster (WCA) (Appendix A). Three samples from Minas Gerais States (2017) were sequenced and clustered together with other strain from the outbreak 2016–2017 into SAI group (A) and some remained ungrouped (B, C, D and E). The Venezuelan strains clustered together (Ven) except INHRR 1AP at 2004 that was included into the group of Trinidad and Tobago (C). Four strains sequenced in this study were confirmed as being vaccine strains: MIS1034 and IQD8393 (from Peru), P16065 (from an imported case from the USA) and OBS5026 (from Ecuador). Two YFV strains from Peru (strains IQT5591 and OBS2240) were characterized as belonging to SAII genotype and the remaining as SAI genotype (Appendix A).

## 3. Discussion

YF epidemics are considered a real challenge by the public health authorities not only in a local level, but also from an international perspective [19]. Due to expansion of YFV circulation in Latin America and Africa, several countries have included the YFV vaccine on their regular immunization programs, or conducted vaccination campaigns [13].

In Brazil, sporadic cases of YFV infections were reported in humans and NHP, specially in the North and Midwest regions [4]. Historically, mass YFV vaccination programs were restricted to these areas, due to evidence of viral circulation [13]. However, several factors have contributed to the reemergence and dispersal of YFV to areas previously free of the disease which then led to its spread in early 2000s to the Southeast regions.

One of these factors was that despite the fact the 17D live attenuated vaccine has been used since 1942 in Brazil, when the last urban transmission cycle was reported [20,21], there was no recommendation of vaccination for the Southeast region as this region was not considered at risk of YF. This situation left the populations living in these areas highly susceptible to YFV [13], leading the Southeast region of Brazil to be at higher risk of YFV transmission.

Between 2016–2018, this scenario culminated in the largest epidemic and epizootic episode of the Americas in the last 80 years, with more than 2153 human cases with 744 fatal cases and 2276 NHP epizootics confirmed [22,23]. The highest number of cases occurred in Minas Gerais (604; 28%) and São Paulo (397; 18.5%).

During the outbreak, the main concern was that most municipalities of this area were unable to reach a vaccine coverage rate of >80% (optimally >95%) [24] to prevent a catastrophic scenario: the yellow fever re-urbanization. Then, a mass vaccination campaign started in the early days of January 2017, using a fractionated dose of the 17DD vaccine [23,25].

Through the whole of the 2016–2017 outbreak the Brazilian government has distributed 26 million extra YFV vaccine doses to affected areas in the states of Southeast region—SP, MG, RJ, and ES [11]. Coverage of >95% was reached in 192 municipalities, 75–95% in 381 municipalities, <75% and >50% in 477 municipalities, and <50% in 126 municipalities [16]. However, the vaccine campaign also brought concern on the occurrence of severe YFV vaccine adverse effects. 

In the past, neurotropic side effects were noted during mass vaccination campaigns [26], and variations in the immunogenicity as well. Furthermore, several cases of viscerotropic adverse effects with dissemination of YFV 17D vaccine were reported, with high lethality rate, in the United States and Brazil [18,27,28]. For this reason, we developed the TaqMan™ system duplex RT-qPCR in order to investigate vaccine adverse effects during YFV vaccination campaigns. Our protocol showed higher sensitivity and specificity than the other conventional protocols. In addition, the protocol is easy to standardize [29,30], which will optimize the diagnostics response time for patients in medical assistance during epidemics. 

The linearity of the duplex was established for the range 10^7^ to 10^3^ copies/reaction (R^2^ 0.99), and this interval may be considered as the reportable range of our assay. The duplex presented an excellent analytical sensitivity with LOD similar to singleplex for both targets (10^2^ copies/reaction), however for quantification we assume the LOQ of 10^3^ copies/reaction. These results were in accordance with the lowest possible concentration that the method is capable to accurate detect in all tested replicates within the reportable range, as recommended by ANVISA (Brazilian Regulatory Agency) [31]. Furthermore, EF% values for YFV RNA amplification were very similar in singleplex and duplex formats, ensuring the same performance for both protocols, and higher overall metrological quality [32,33], especially with low target amounts [34,35].

The RNAse P gene has been used as an endogenous internal control (End-IAC) in different protocols for human samples [36] and eventually to NHP, while the MS2 RNA was used as exogenous amplification control for mosquitoes and some NHP samples [37]. Furthermore, the RT-qPCR cycling conditions of our assay are compatible with the protocols for the detection of other arbovirus genomes, such as DENV [38], ZIKV [39], CHIKV [40], WNV [41], Mayaro virus (MAYV) and Oropouche virus (OROV) [42] allowing then to be performed simultaneously in the same plate. This is a remarkable feature, and is very important for diagnostics optimization and for to the surveillance of other arboviruses during outbreaks.

For analytical specificity analysis, we checked two parameters. First we verified the ability of duplex to detect other arboviruses genome that are genetically or clinically related to YFV. Second, we have tested YFV samples from different South American countries. The negative amplification of other 25 viruses and 100% of YFV strains detection indicates that this assay had the highest (100%) specificity to detect YFV genome, including strains from different YFV genotypes (I and II) as well as strains from different geographic regions, ecosystems and years of isolation. 

It is important to note that our 17DD primers/probe alone are not suitable to distinguish the vaccine strain from wild-type strains from the Eastern and Central African genotype. Viral genome sequencing is necessary to confirm a potential YEL case. In addition, genome sequencing would be of great relevance for surveillance, as a tool to detect mutations in 17DD strains associated with YEL as well as to detect African strain in case of introduction in Brazil. Other real time protocols had been established to differentiate YFV wild infection from vaccine, but most of them were not able to differentiate vaccine from Africa strain and were not optimized to be used in one single tube [31,43]. The exception is the RT-qPCR CDC-Colorado assay recently published that was able to specifically identify the vaccine strain based on SNPs detection of eight single mutations [44]. It has been reported that 17DD vaccine is quite stable [45,46] and that most of adverse reactions are probably due an individual genetic susceptibility because no mutations have been found in the vaccine virus recovered from these patients. However, observing the magnitude of the last YF outbreak, the implementation of this protocol in the YFV diagnosis network in Brazil was difficult due to multiple primers/probes target required. On the other hand, while serious adverse reactions (YEL-AND and YEL-AVD) caused by YFV 17D vaccine are rare, ranging from 1 per 250,000 to 1 per 500,000 vaccinations [14,28,47], the NGS may be a good and viable strategy to be applied during outbreaks to confirm duplex RTqPCR results.

The duplex RTqPCR performance was also evaluated using 53 clinical samples which had previous results from virus isolation and the conventional RT-PCR method. Our results showed that the conventional RT-PCR assay sensitivity is very similar to the virus isolation technique. However, RT-qPCR showed around 4-fold higher sensitivity when compared to those YFV traditional diagnostics tools. This difference is probably due to the low LOD observed for the conventional RT-PCR, with values about 9 logs lower than the RT-qPCR LOD. Thus, this duplex RTqPCR assay could be a useful diagnostic tool capable of surpassing some of the sensitivity limitations of the other viral detection protocols. 

In Figure 4, an overview of the CT values for each biological sample (humans and PNH), as well as mosquitoes pools is shown. This information could be used as reference in future YF investigation using this protocol. We noticed that liver samples showed the lowest CT values for human and NHP. Due to hepatotropic behavior of the YFV, the liver is the primary target for YFI, being critical to the establishment of the severe classic disease.

YF pathogenesis is viscerotropic in humans, with viral replication in the liver being critical for the establishment of the disease [48]. In this organ, YFV induces hepatocyte apoptosis and lytic necrosis, which, combined with steatosis, results in most of the liver damage observed during infection [48]. Studies in human cells suggest that virulent YFV and/or YFV-17D display(s) a very broad tissue tropism and can replicate in hepatocytes [49,50], various hematopoietically derived cells (including DCs [51,52], monocyte-derived macrophages (MDMs) [53], T cell Kupffer cells [54], and endothelial cells [55]. These data corroborate with our finding that the liver is the main organ affected, showing high rates of viral replication.

In addition, YFV affects other organs, including kidneys, spleen, lungs, pancreas, lymph nodes and heart [48,49], and should not be ignored during the occurrence of severe disease in an outbreak. 

From 70 confirmed cases, 84.3% occurred during or after the mass vaccination campaign in Southeast region. This investigation was requested by the Brazilian Ministry Health to verify the vaccine safety and the eventual side effects. Our data corroborate previous report of low prevalence of YEL induced by 17DD vaccine where just two (2.89%) postvaccine yellow fever cases have been confirmed.

Although no yellow fever vaccine efficacy studies have been conducted, the vaccine is reliably immunogenic worldwide and neutralizing antibodies develop by day 10 after vaccination in 80% of yellow fever vaccines [50]. Primary yellow fever vaccine recipients have self-limited, vaccine-derived, viremia, that typically lasts after 3–4 days post vaccination. However, this postvaccination viremia can last as long as 2 weeks. Thus, the detection of yellow fever viral RNA by RT-PCR testing around 3 days postvaccination or after 13 days needs to be confirmed if caused by natural wild-type infection (acquired either before vaccination or later if there is vaccine failure) or yellow fever vaccine-associated viscerotropic disease (YEL-AVD). A rare, but serious AEFI occurs when the vaccine-derived virus proliferates in multiple organs after primary vaccination. The two YEL-AVD cases confirmed in this study and that were detected during the vaccination campaign in 2016–2017 had this clinical presentation. The 17DD genome was detected by duplex RT-qPCR 2–3 days after the patients received the vaccine. Despite the earlier detection, analysis of NGS data showed only 17DD genome and no YFV wild type genome.

The symptoms of YEL-AVD are similar to those of naturally acquired yellow fever, typically the disease onset between 2 to 10 days postvaccination, but there are reports that the vaccine-derived viremia can persist beyond 13 days [51,52].

In addition, an example of this was the recent yellow fever epidemic in African countries in 2016, when robust efforts of WHO and partners were needed to support local governments to control the outbreak, to strengthen measures to prevent new cases, and to avoid its spread to other countries. There are essential strategies focused on surveillance and risk assessment, vaccination, case management, vector control, social mobilization and risk communication [19,52].

In summary, the recent YF outbreaks in Latin America [53,54] and Africa [55] associated with massive vaccination campaigns highlights the great necessity of rapid and specific YFV diagnostics tools as part of a global effort to control the disease and to track the AEFI. In this regard, the duplex RT-qPCR developed in this study is a fast, high sensitive and specific assay that may become a powerful diagnostic tool to assist epidemiological investigations and decision-makers during outbreaks or epidemics of YF in South America

## 4. Materials and Methods

### 4.1. Ethical Statement 

Human, mosquitoes and NHP samples for diagnosis and investigation from YF suspected cases, were received at the Department of Arbovirology and Hemorrhagic Fevers, in the Evandro Chagas Institute (SAARB/IEC), Pará State, Brazil National Reference Laboratory for the Brazilian Ministry of Health. Human samples were received accompanied by investigation records containing the patient’s demographic and clinical information. Patients personal information was previously anonymized before use of the data.

### 4.2. Duplex TaqMan RT-qPCR Assay Establishment

A duplex TaqMan RT-qPCR assay was established to distinguish wildtype YFV from the 17DD vaccine strain, by using probes labeled with distinct fluorophores that allowed a simultaneous detection of both targets in a single tube. We used the YFall primers and FAM/3′BHQ1 probe as described by Domingo et al. [18] which target both (wild type and vaccine strains) and combined to primers and probe HEX™/3′BHQ1, named 17DD, targeting the vaccine 17DD (African genotype) strains, designed based on available vaccine strains sequences (Genbank: DQ100292, X03700, DQ118157, YFU17067, GQ379163, GQ379162, YFU17066, YFU21055 and YFU21056) using the Geneious v.9.1.6 software (Biomatters, Auckland, New Zealand). This is presented in Table 3.

Newly designed primers and probes were checked for hairpin secondary structure formation and melting temperature (Tm) compatibility using Geneious and Mfold software. Primer dimer analysis were performed using the AutoDimer software [56].

YFVAll primers and probe has been used in our reference laboratory as singleplex, while the 17DD/African primers and probe were designed for this work. First, we optimized primers and probe concentration for duplex format, showing close to the maximum limit recommended with lower Ct value (primer range from 50–1000 nM; probe range from 50–300 nM). This condition provided early detection of the target, quantification of low target amounts without impairing reaction specificity since nonspecific amplifications were not observed [30,57].

Viral RNA was extracted using QIAamp^®^ Viral RNA Mini kit (Qiagen, Germantown, Germany) or TRIzol^®^ Plus RNA Purification Kit (Thermo Fisher Scientific, Waltham, MA USA). Quantitec Probe RT-qPCR Master Mix (Qiagen) was used for RT-qPCR amplification in the ABI 7500 fast real-time PCR system (Applied Biosystems, Waltham, MA, USA). Thermocycling conditions consisted of one cycle at 50 °C for 30 min for reverse transcription, one cycle at 95 °C for 15min for RT inactivation/initial denaturation, followed by 45 cycles of 94 °C for 15 s and 60 °C for 60 s. Extracted RNA (5 μL) was used as a template in a 20 μL reaction. Samples and controls were run in duplicate. The RNAse P [36] gene was used as endogenous internal amplification control for humans and MS2 for mosquitos and NHP samples [58,59] in separate single tubes.

For the assay analytical sensitivity analysis, an in vitro transcribed (IVT) RNA containing the target YFV sequence was generated to determine the limits of detection (LOD), limits of quantification (LOQ), linear dynamic range (LDR) and linear correlation coefficient (R^2^). EF% was calculated by the standard curve method [60] using a 10-fold, 7-log, serial dilution starting at 2 × 10^7^ RNA copies/μL, in duplicate, and performance was compared between singleplex and duplex format. YFV IVT RNA was transcribed using Megascript T7 or SP6 (Thermo Fisher Scientific, Waltham, MA, USA), purified using the Megaclear kit (Thermo Fisher Scientific, Waltham, MA, USA) according to the manufacturer instructions and quantified using the Qubit RNA BR Assay kit (Invitrogen, Waltham, MA, USA) in the Qubit 2.0 digital fluorimeter (Invitrogen). 

Primers and probes analytical specificity was evaluated using extracted RNA from the YFV wild type strain (BeH111), from the YFV vaccine strain (17DD) and 25 viruses from the SAARB/IEC’s collection and representative of the following families: Flaviviridae (Dengue virus serotypes 1 to 4, Zika virus, Ilheus virus, Saint Louis Encephalitis virus, Rocio virus), Orthobunyaviridae (Catu virus, Caraparu virus, Tacaiuma virus, Icoaraci virus, Utinga virus, Oropouche virus and Jatobal virus), Hantaviridae (Laguna Negra virus) and Togaviridae (Chikungunya virus, Mayaro virus, East Equine Encephalitis virus, West Equine Encephalitis virus, Pixuna virus and Mucambo virus). All strains were previously isolated in newborn Swiss mice brain and the viral RNA was extracted as described previously. For the RT-qPCR assay, 20 ng of extracted RNA was used.

Furthermore, we tested YFV strains (*n* = 28) representative of all genotypes/strains circulating in Latin American countries (Brazil, Ecuador, Venezuela, Peru, and Trinidad & Tobago) and USA, from 1968 to 2017, kindly provided by the University of Texas Medical Branch (UTMB, Galveston, TX, USA (Appendix A).

### 4.3. Duplex TaqMan RT-qPCR Assay Evaluation

For the assay evaluation and for comparison purposes, we additionally submitted clinical samples (*n* = 53) from 30 YF suspected cases from the Brazilian outbreaks 2016–2017, including two acute YEL-AND cases defined according to the Brazilian Ministry of Health (MoH) criteria and with less than 5 days of symptoms to virus isolation in Vero cells, to the conventional RT-PCR and to the Duplex RT-qPCR assay established here. 

### 4.4. Investigation of YF Cases, Vaccine Adverse Events, Vectors and Epizootics during the 2016–2017 Epidemic

The Duplex RT-qPCR was further evaluated during the 2016–2017 YF epidemic in Brazil. A total of 489 clinical samples from 319 human cases, including serum (*n* = 180), blood (*n* = 12), brain (*n* = 19), liver (*n* = 117), spleen (*n* = 72), pool of tissues (lung, heart and kidney) (*n* = 83) and cerebrospinal fluid (CSF) (*n* = 6) from the states of Pará, Acre, Amazonas, Bahia, Federal District (Brasília), Espírito Santo, Goiás, Maranhão, Minas Gerais, Mato Grosso, Paraíba, Piauí, and Rio de Janeiro were tested. All tissue samples used in our study to confirm the diagnosis are postmortem samples. Aiming to evaluate its usefulness for epizootic studies, a total of 949 clinical samples from 512 NHP cases, including serum (*n* = 46), blood (*n* = 43), brain (*n* = 82), liver (*n* = 315), spleen (*n* = 339) and pool of other viscera tissues including heart and kidney (*n* = 124) were tested. The assay usefulness was, still, evaluated for entomological surveillance, where 41 pools of mosquitoes captured in Minas Gerais was tested. The specimens were identified as *Haemagogus sp*. (*n* = 31) and *Sabethes sp*. (*n* = 10). This replicated in Table 2.

### 4.5. Next Generation Sequencing

To confirm the Duplex RT-qPCR results, 31 samples were sequenced using the 454 GS FLX System (Roche, Basel, Switzerland) and MiniSeq (Illumina Inc, San Diego, CA, USA) platforms (Appendix A). The first and second strand RNA synthesis was performed using SuperScript VILO MasterMix (Invitrogen) and NEBNext Second Strand Synthesis Module (New England BioLabs Inc, Hitchin, UK), respectively. The reaction was purified with PureLink PCR Purification Kit (Invitrogen). The cDNA library was generated using the cDNA Rapid Library Preparation Method Manual GS FLX+/XL to 454 and Nextera XT DNA Library Preparation Kit to MiniSeq. The genome sequences were determined using the De Novo Assembler methodology in IDBA-UD program [61] and SPAdes [62]. All contigs were aligned and compared to the YFV genome database available in NCBI through Diamond [63]. Inspection, annotations of putative ORF/genes and additional analysis were performed using the Geneious v.9.1.6 software (Biomatters). 

A Multiple Sequencing Alignment (MSA) was performed by Mafft v7.310 software [64], spanning the entire Brazilian strains ORFs and 30 YFV sequences available on NCBI GenBank. Recombination events were evaluated using Phi-test implemented on SplitsTree4 v4.14.6 program [65]. The best-fitting model of nucleotide substitution was determined using jModelTest v.2.1.10 [66]. The phylogenetic inference was performed using the Maximum Likelihood (ML) methodology by RAxML v.8.2.11 software [67]. To facilitate understanding, we have developed a study flowchart that is replicated in Figure 5.

## 5. Conclusions

The new RT-qPCR protocol is able to detect YFV infections caused by both wild type or vaccine strains with high specificity and sensitivity and could become a valuable tool for reference laboratories. In our hands, this assay achieves an excellent diagnostic performance on clinical samples from humans, NHP and mosquitoes and notably facilitates detection of YFV in different types of samples. In addition, the diagnostic assay was extensively validated using more than 1400 clinical samples obtained from different sources, presenting an excellent performance.

The YF RT-qPCR assay demonstrated broad-spectrum detection of vaccine YFV, since it allowed for the detection of two cases of YFV-AVD during the 2016–2017 epidemic, evidencing the usefulness of our duplex RT-qPCR to identify these cases that will probably occur due to the expansion of the 17DD vaccination area in Brazil.

Our results demonstrated that the type of sample most suitable for the molecular diagnosis of YFV is the liver, because, this tissue should present lower Ct values, allowing a better detection of YFV in this type of sample. It is worth mentioning that if available, other tissues should also be investigated since other organs are also affected during the disease.

Through our findings, while continued validation is warranted, the RT-qPCR duplex assay can be used as a rapid and feasible clinical diagnostic tool during YFV outbreaks globally to assist epidemiological investigations and decision-makers during mass vaccination campaigns in South America

## Figures and Tables

**Figure 1 pathogens-10-00693-f001:**
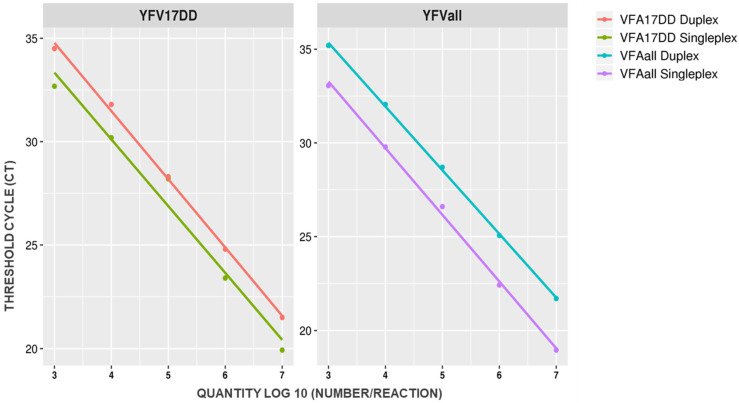
Comparison of primer performance in singleplex and duplex RT-qPCR formats.

**Figure 2 pathogens-10-00693-f002:**
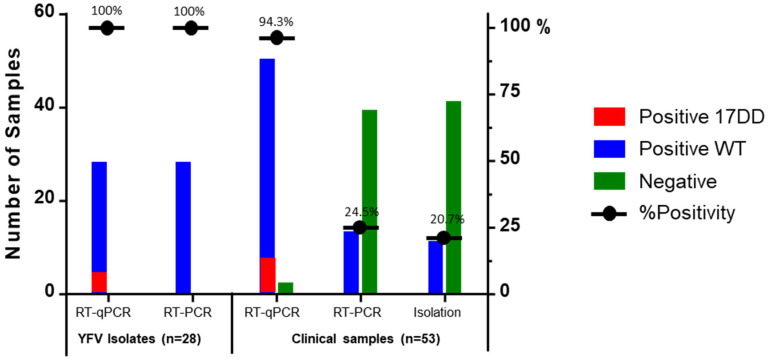
Evaluation of the Duplex RT-qPCR in YFV isolates (*n* = 28) and YF suspected cases (*n* = 53) in comparison to methods routinely used for diagnosis (isolation and RT-PCR).

**Figure 3 pathogens-10-00693-f003:**
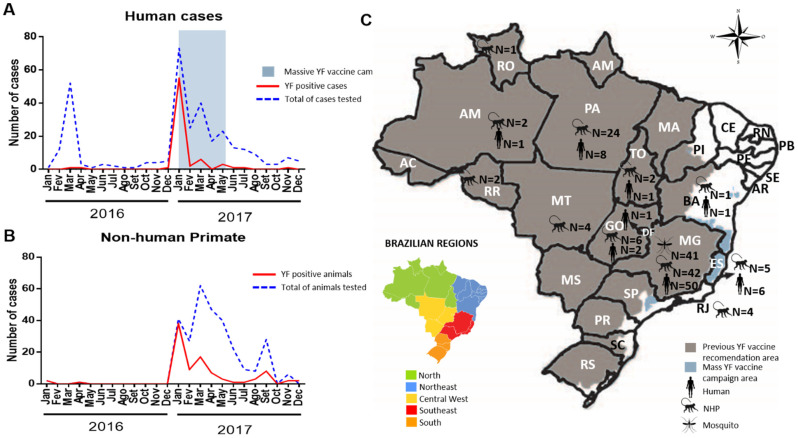
Distribution of confirmed yellow fever human cases (*n* = 70/319) and NHP (*n* = 94/512) investigated during the 2016–2017 epidemic in Brazil. Monthly distribution of human (**A**) and NHP cases (**B**). Distribution of human cases as well as NHP cases and mosquitoes positive for YFV per state (**C**).

**Figure 4 pathogens-10-00693-f004:**
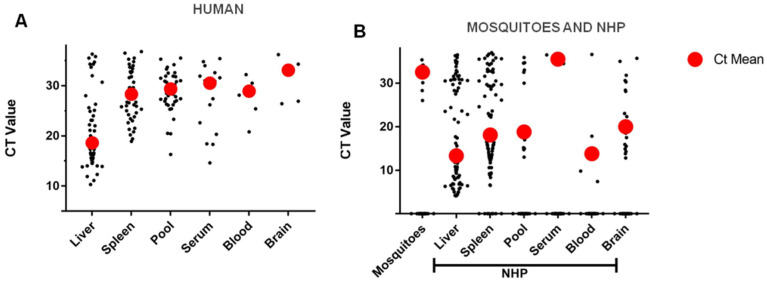
Distribution of the Ct values of the positive samples used in the article. (**A**) human samples (**B**) samples of NHP and mosquitoes and the red circle shows the mean Ct.

**Figure 5 pathogens-10-00693-f005:**
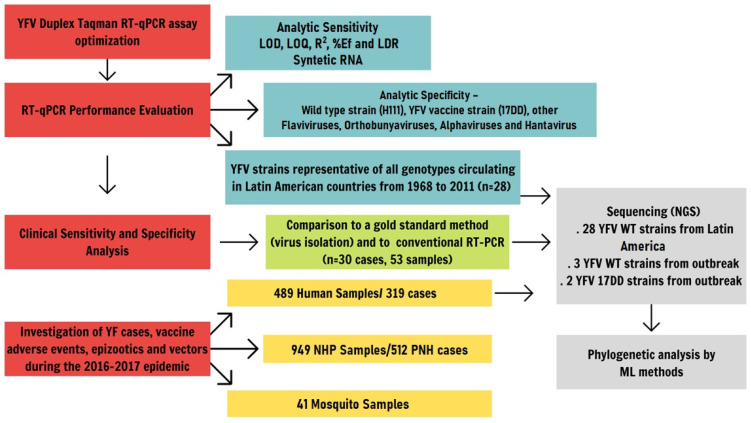
Study flowchart to evaluate the usefulness of a Duplex RT-qPCR for the surveillance of vaccine adverse events, epizootics and vectors during the 2016–2017 yellow fever Brazilian epidemic.

**Table 1 pathogens-10-00693-t001:** Parameters obtained in both formats (Ct = cycle threshold; ND: not detected; QL = Quantification Limit; DL: Detection Limit; EF% = Efficacy; R^2^ = linearity coefficient; LOD = Limit of detection; LOQ = Limit of quantification; LDR = Linear Dynamic Range).

PARAMETERS	SINGLEPLEX		DUPLEX	
	YFVall	YFV17DD	YFVall	YFV17DD
Ct cut off	<38	<38	<38	<38
Slope	−3.359	−3.226	−3.314	−3.304
Y-inter	42.856	46.86	45.039	44.778
R^2^	0.993	0.993	0.997	0.997
EF%	98.48%	104.1%	100.34%	100.76%
LDR	10^3^ to 10^7^ copies/reaction		10^3^ to 10^7^ copies/reaction	
LOD	10^2^ copies/reaction		10^2^ copies/reaction	
LOQ	10^3^ copies/reaction		10^2^ copies/reaction	

**Table 2 pathogens-10-00693-t002:** Percentage of positivity to YF in human, NHP and vectors samples.

Host	Specimen	Positive (%)
Human	Serum (*n* = 180)	8.89
	Blood (*n* = 12)	50.00
	Liver (*n* = 117)	45.30
	Spleen (*n* = 72)	69.44
	Brain (*n* = 19)	26.32
	Pool (*n* = 83)	48.19
	Cerebrospinal Fluid (*n* = 6)	0
	TOTAL = 489	
NHP	Serum (*n* = 46)	4.35
	Blood (*n* = 43)	6.98
	Liver (*n* = 315)	26.98
	Spleen (*n* = 339)	22.71
	Brain (*n* = 82)	26.83
	Pool (*n* = 124)	10.48
	TOTAL = 949	
Mosquitoes	*Haemagogus* spp. (*n* = 31)	19.51
	*Sabethes* spp. (*n* = 10)	0
	TOTAL = 41	

NHP—Non-Human Primates.

**Table 3 pathogens-10-00693-t003:** Primers and probe sequences used for the duplex qRT-PCR for YF surveillance and vaccine adverse event determination.

Primers	Sequence (5′→3′)	Genome Position ^b^
YFALL R	5′- CTG CTA ATC GCT CAA MGA ACG -3′	83–103
YFALL F	5′- GCT AAT TGA GGT GYA TTG GTC TGC -3′	15–38
VFA 17D R	5′- TTT AAG TGC GGA GYC CGG TT- 3′	10448–10667
VFA 17D F	5′- TAC AAA CCA CGG GTG GAG AA -3′	10382–10401
Probe	Sequence (5′→3′) ^a^	Genome Position ^b^
YFALL	5′-FAM-ATC GAG TTG/ ZEN/ CTA GGC AAT AAA CAC-BHQ1 -3′	41–64
VFA 17D	5′-HEX-ACT TGA AAC/ ZEN/ CGG GAT ATA AAC CAC GGC TGG-BHQ1 -3′	10416–10445

^a^ FAM, 6-carboxyfluorescein; HEX, hexachloro-6carboxy-fluorescine; ZEN, internal quencher. ^b^ Positions are indicated relative to GenBank sequence AY640589. 1 for Yellow Fever virus Asibi strain.

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
