# Peer review of "The Usefulness of a Duplex RT-qPCR during the Recent Yellow Fever Brazilian Epidemic: Surveillance of Vaccine Adverse Events, Epizootics and Vectors"

_pathogens, 2021, doi:10.3390/pathogens10060693_

Round 1
Reviewer 1 Report
Queiroz et al. present a manuscript describing the use of a Duplex RT-qPCR during the latest yellow fever outbreaks in Brazil. The analysis included surveillance of vaccine presence and adverse effects in addition to the presence of YFV in non-human primates and mosquito vectors. The authors develop their own qPCR protocol, they validate it and test is against several YFV strains including the vaccine showing specificity to those strains.
The manuscript is enjoyable to read, we researched and interesting. It has a good flow. Materials and methods are appropriate, sample size sufficient and research seems methodical. The results and figures are clear and the analysis is good. The conclusions are also well funded.
The topic of vaccine adverse effects is quite interesting, especially lately. The authors explain it well and have sound conclusions. This research is very useful. The development of this Duplex qRT-PCR seems a great tool for future outbreaks to distinguish between vaccinated vs non vaccinated cases.
Overall the paper is very good.
Author Response
Dear Reviewer 1,
In the name of all the authors, I would like to thank you very much for the review of our manuscript entitled: “The usefulness of a Duplex RT-qPCR during the recent yellow fever Brazilian epidemic: surveillance of vaccine adverse events, epizootics and vectors”. We are grateful for your consideration of this manuscript, and we also very much appreciate your suggestions, which have been extremely helpful in improving the manuscript. Finally, we also want to thank the Reviewer for their important comments and specially for the classification of the manuscript as very good.
Sincerely,
Alice Queiroz
Reviewer 2 Report
In their manuscript, Queiroz et al describe the design and implementation of a novel real time PCR assay for the detection of both wild-type and vaccine related Yellow fever infection. The assay was evaluated against a panel of 28 strains of Yellow fever virus from Latin American countries over a 40-year range, and critically, is able to differentiate wild-type YFV infection (South American phylogenetic cluster) from vaccine associated (West Central African cluster/Asibi parental) signal making this a invaluable resource in South America. Given the ongoing sylvatic circulation of YFV in South America, the potential for epizootic spillover and subsequent outbreaks is pronounced therefore the presented assay has enormous utility, especially when evaluated against the current gold standard of virus isolation. The present study is well designed and conducted soundly. A few minor concerns are noted below.
- (Throughout): The manuscript does need editing for clarity, syntax, word choice and sentence structure throughout.
- Figure 1: The left axis label (Threshold Cycle (CT)) on the left chart is impeding the view of the actual axis values (20, 25, 30 CTs)
- In Figure 2: The number of samples in the right half of the graph is 52. In the text (Lines 134-139), 53 samples are referred to. This discrepancy is also present in the percentages in the text vs the figure.
- Given the emphasis of the value of liver samples for diagnosis, the authors should a)specify the nature of these samples (biopsy? Post-mortem?), and discuss why this standard of sample is so critical for diagnostics.
Reviewer 3 Report
In this manuscript by Queiroz et al. the authors describe the development of a duplex qRT-PCR assay to detect and distinguish between South American, African and vaccine strains of Yellow fever virus (YFV). The authors then employ this assay to confirm virus infections in humans, non-human primates and mosquitoes in the 2016/17 YF epidemic in Brazil with particular emphasis on the identification of potential vaccine-induced adverse events.
This manuscript has potential and I can appreciate the effort made but it requires a major revision prior to publication. In particular I would suggest the authors consider the following points:
- With the exception of the Materials and Methods section the English and the punctuation need to be corrected. Nearly every sentence has spelling mistakes or is grammatically incorrect. It made reading the manuscript more difficult than it needed to be. I am happy to provide a copy of the manuscript in which I have highlighted some of these mistakes.
- There was no rationale given for why the detection of vaccine-induce adverse events was of importance, especially during a large outbreak in an area with suboptimal vaccine coverage where one would expect that most cases are virus-induced and given the low reported rates of these events.
- There was no introduction in the results section to inform the reader of how the duplex qRT-PCR assay had been designed and optimised. Abbreviations are used without any explanation which made it impossible to follow the narrative. It was only when I read the materials and methods section that the results section made sense. I would suggest to use part of the materials and methods section instead of the results section, especially given that the English is also a lot better in the methods. Figure 1 has no legend and the y-axis scale for the 17D panel is covered by the title. Table 1: Should "multiplex" not be "duplex"? It is unclear what the sentence in lines 107 and 108 means. Figure 2: It is unclear which are the "routinely used methods". Line 124: There is no introduction as to why NGS was performed or which samples were used. Again, this is much better described in the materials and methods. Thus, I suggest a restructuring of the text. Figure 3: the sub panels are not labelled. What do the red and blue lines in panels A and B represent? In panel C is should be "mass YF vaccination campaign area". Table 2: I would prefer "Positives (%)" instead of "Positive/Tested (%)". Figure 4: y-axis scales inconsistent in panels A and B. The figure legend has too little detail. Lines 195-202: Absolutely no explanation or details are given. What is an ML reconstruction and what does GTR+I+G mean?
- Line 228 mentions 36.7 million vaccine doses for the SE region, while line 77 mentions 26 million.
- Line 79: 14.5% of 100.000 doses would be 14.500 cases of adverse vaccine events which cannot be correct.
- Lines 242-248: this should go into materials and methods
- Line 259: What does IAC stand for?
- Line 293: Can the authors please use one style for duplex/Duplex qRT-PCR/RTqPCR/RT-qPCR throughout the manuscript.
- Figure 5: please correct spelling of analytic and synthetic.
Please note I had no access to the supplementary data file and have thus not commented on the figures and tables therein.

Round 2
Reviewer 3 Report
I would like to thank the authors for carefully addressing a majority of my previous comments. I believe that the manuscript has indeed significantly improved.
With regards to the introduction of methods in the results section, I would still suggest that section 2.1 begins with a sentence introducing the rationale for the work that has been conducted. Something simple along the lines of: "In this study we aimed to develop a duplex qRT-PCR assay able to detect and distinguish between vaccine and wildtype strains of YFV in Brazil. For this, we designed primer and probe pairs against xxx YFV strains and optimised primer and probe concentrations. We found that our assay was able to detect xxx YFV strains with xxx efficiency... ."
This would serve as a short introduction to the methods without going into any details. Details such as primer and probe concentrations used and cycling conditions can then be described extensively in Methods. I agree that a manuscript has distinct sections, obviously, however, all sections need to be connected by a few sentences for the benefit of the reader. Without having read the Methods section at the end of the manuscript I should still be able to follow the Results section.
There are still some spelling and grammar mistakes that should be addressed.
